# Partial-labeled Abdominal Organ and Cancer Segmentation via Cascaded Dual-decoding U-Net

Zhiyu Ye[1,2,3][0009−0001−1520−0051], Hairong Zheng[1,3][0000−0002−8558−5102], and Tong Zhang[2][0000−0002−8838−4963]

[1] Shenzhen Institute of Advanced Technology, Shenzhen, China
[2] Peng Cheng Laboratory, Shenzhen, China
[3] University of Chinese Academy of Sciences, China
zhangt02@pcl.ac.cn

**Abstract.** In the FLARE2023 challenge, we developed a cascaded dual-decoding U-Net framework to address the complex task of partial-labeled abdominal organ and cancer segmentation. Initially, we explored the potential of 3D transformer-based models but transitioned to 2D U-Net solutions due to computational resource and inference time constraints. We first trained separate 3D models for cancer and full-organ segmentation using data that included labels for both cancer and full organs. Subsequently, we generated pseudo labels for unlabeled and partially labeled data based on these initial models. To enable a single model to effectively learn and infer both organ and cancer labels within images, we designed a dual-decoding structure based on the 2D U-Net architecture. Our training process involved several steps with various subsets of the training data. By comparing our model trained without unlabeled data, we discussed the impact of unlabeled data and its pseudo labels on the experimental results. Our method, the version trained without unlabeled data, achieved an average DSC score of 83.22% for organs and 33.22% for lesions on the validation set. The average running time and area under the GPU memory-time curve were 33.8 seconds and 50066.25MB, respectively. The codes has been open-sourced to https://openi.pcl.ac.cn/OpenMedIA/pclmedia_FLARE23.

**Keywords:** FLARE2023 · Partial-labeled abdominal organ and cancer segmentation · U-Net.

## 1 Introduction

The segmentation of abdominal organs and lesions has always been a classic research task in medical image analysis and also plays a fundamental role in facilitating medical practitioners in areas such as diagnosis, surgical planning, and various clinical applications. In open datasets and challenges focused on individual organs, such as LiTS [2] and KiTS [11,12], the developed models have consistently achieved impressive results, with state-of-the-art Dice Similarity Coefficient (DSC) scores consistently surpassing 0.95. Furthermore, researchers are

also working towards the development of models with the capacity to concurrently segment multiple organs and lesions, ultimately augmenting the utility of automated segmentation in medical practice. However, the existing open datasets for multiple abdominal organs often fall short of meeting the demands for training comprehensive segmentation models encompassing all abdominal organs and lesions. To illustrate, the AbdomenCT-1K dataset [21] covers only four abdominal organs, while the BTCV dataset [15], though has thirteen organ labels, comprises only 50 images, and none of these two datasets includes lesion segmentation.

Addressing the time-consuming and labour-intensive nature of labelling targets for segmentation in extensive medical images is a significant challenge that must be tackled in the development of medical image segmentation algorithms. A viable solution is trying to make the most of labeled data for certain organs and lesions, or even leveraging unlabeled data. From this perspective, FLARE2023 offers a dataset comprising 1800 unlabeled and 2200 partially labelled CT images, with the aim of encouraging participants to develop solutions that can effectively perform simultaneous segmentation of thirteen abdominal organs and cancer.

In recent years, extensive research has been conducted to address the problem of partial-label segmentation for abdominal organs and cancer. Several methods have emerged as promising solutions for this task. One approach is to achieve dynamic and diverse object segmentation by incorporating with adaptive filters during the decoding or output stages within a unified encoding-decoding architecture. Notable examples of this approach include DoDNet [31] and the conditional nnU-Net [30]. Another prevalent architectural design tailored to this domain is the implementation of a multi-head decoder. For instance, models like MFUnetr [8] incorporate separate segmentation heads for both full and partial organ segmentation. In terms of learning strategies, researchers have explored various methods to utilize the unlabeled data. These strategies encompass multi-stage training, model distillation and integration, semi-supervised learning method such as pseudo label generation, sometimes even combining multiple strategies. It is noteworthy that the FLARE2022 conference proceedings [17] feature an extensive array of solutions that exemplify the practical application of these methods.

Moreover, self-supervised learning methods offer a practical approach to this task. One widely used strategy is the pre-training and fine-tuning of transformer-based networks. During the pre-training stage, an abundance of unlabeled images can be leveraged to equip the transformer encoder with the ability to comprehend input images and extract meaningful features. An illustrative example is Swin UNETR [26], which achieved state-of-the-art results on BTCV [15] and MSD [1] datasets after pre-training on 5050 unlabeled data. Similarly, UNETR [9], which also employs a vision transformer as the encoder, can employ this self-supervised learning approach.

At the outset, we embarked on a self-supervised learning approach with 3D transformer-based networks, specifically Swin UNETR and UNETR. In parallel,

we supervised trained a 3D U-Net model using a limited portion of labelled data for comparison. However, it became evident that the GPU memory consumption of 3D models far exceeded the specified 4GB. As a result, we pivoted directly to a different strategy, opting for the classic 2D UNet architecture and embracing a semi-supervised training strategy. This involved initial training on unlabeled data with pseudo labels generated by 3D U-Net, and the model was sequentially trained on data with varying label patterns. Ultimately, this revised approach yielded a model that surpassed our initial 3D models in terms of both segmentation performance and inference efficiency. In this paper, we will discuss our solution and contributions from the following aspects:

- We adopted various strategies and trained multiple networks to address this task, including transformer-based Swin UNETR and UNETR, as well as CNN-based 3D and 2D U-Net models for this task. We conducted comprehensive comparisons and in-depth analyses of the results derived from these varied approaches.
- To facilitate training on partially-labeled data, we devised a dual decoding structure based on the 2D U-Net architecture. This design enables us to fix certain parameters while updating others during different training steps.
- In our comparison between 2D models trained with and without unlabeled data, we observed that even the model was trained with inaccurate pseudo labels of unlabeled data, it led to an improvement in the model's performance.

## 2    Method

Fig. 1 presents an overview of our method. Given the evolution of our method, transitioning from a transformer-based to a CNN-based approach and from 3D to 2D, it is structured into two distinct stages: the 3D model training stage (Fig. 1(b)) and the 2D model training stage (Fig. 1(d)). Within each training stage, several steps were undertaken, and each step is trained on a subset as defined in Fig. 1(a) of the training data. Notably, the 3D models play a pivotal role in generating pseudo labels for the subsequent training of the 2D model, as illustrated in Fig. 1(c).

### 2.1    Data Partition.

There are a total of fourteen classes to be segmented in the FLARE2023 task, where labels 1 to 13 correspond to thirteen abdominal organs, and label 14 corresponds to cancer. The training data can be categorized into different subsets based on their label patterns, as demonstrated in Figure 1(a):

- $D_u$: 1800 unlabeled images.
- $D_{l1}$: 250 images with labels for all thirteen organs (labels 1 to 13).
- $D_{l2}$: 458 images with labels for only five organs (labels 1, 2, 3, 4, and 13), representing the liver, right kidney, spleen, pancreas, and left kidney, respectively.

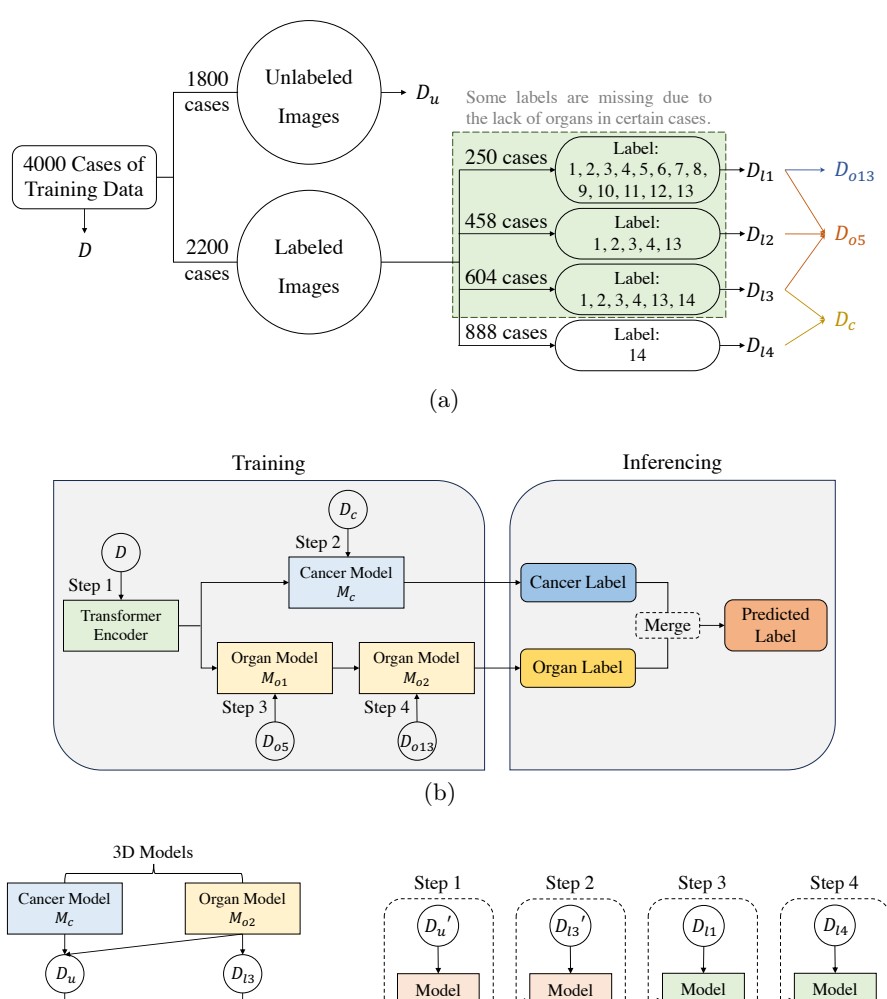

**Fig. 1.** Visualization of our data partitioning method and training strategy. (a) Data Partitioning. The entire training dataset $D$ has been segmented into five distinct subsets $D_u, D_{l1}, D_{l2}, D_{l3}$, and $D_{l4}$. Specifically, $D_{o13}, D_{o5}$, and $D_c$ represent data containing 13 organ labels, 5 organ labels, and cancer labels, respectively. (b) Training and Inference Process for 3D Models. Transformer-based models underwent four training steps, while the 3D U-Net skipped the first two steps. (c) Pseudo Label Generation with 3D Models. The trained 3D U-Net $M_c$ and $M_{o2}$ are employed to generate pseudo labels for cancer and organs in images from datasets $D_u$ and $D_{l3}$. These images with pseudo labels are denoted as $D'_u$ and $D'_{l3}$. (d) Training Process for 2D U-Net. In the first two training steps, all model parameters were updated, while in steps 3 and 4, only some of the model parameters were updated.

- $D_{l3}$: 604 images with labels for the same five organs as $D_{l2}$, as well as cancer (label 1, 2, 3, 4, 13 and 14).
- $D_{l4}$: 888 images with label exclusively for cancer (label 14).

Therefore, the complete training dataset $D$ is the union of $D_u$, $D_{l1}$, $D_{l2}$, $D_{l3}$, and $D_{l4}$, with no overlap between these subsets. Additionally, for training purposes, we denote $D_{l1}$ as $D_{o13}$, the union of $D_{l1}$, $D_{l2}$, and $D_{l3}$ as $D_{o5}$, and the union of $D_{l3}$ and $D_{l4}$ as $D_c$. These subsets represent data containing thirteen organ labels, five organ labels, and cancer labels, respectively. It's worth noting that within $D_{o5}$ and $D_{o13}$ subsets, not all images have labels as described above. There are instances where certain organs are missing, resulting in some images lacking one or several labels corresponding to the absent organs.

### 2.2 Pre-processing

For the 3D models, we applied uniform pre-processing to ensure consistent orientation, intensity range, and spacing for all input images:

- Orientation: The orientations of 3D images were standardized using the 'RAS' axcodes.
- Scale Intensity Range: The values of image voxels were clipped to $[-200, 300]$ and then normalized to $[0, 1]$.
- Spacing: The spacings were resampled to a uniform spacing of $1mm \times 1mm \times 1mm$ using bilinear interpolation for images and nearest neighbour interpolation for labels.

For the 2D models, the pre-processing steps were similar but with some parameter differences. Specifically, the values of image voxels were clipped to $[-200, 300]$ when scaling the intensity ranges. Besides, in the training stage, images were resampled to the spacing of $1mm \times 1mm$ on the height and width dimensions, while the depth dimension remained unchanged. However, during inference, images were resampled to the spacing of $1mm \times 1mm \times 2.5mm$. This adjustment aimed to prevent long inference times caused by some images with small spacing and an excessive number of slices. After these transformations, the images were sliced into 2D samples along the depth dimension to prepare for training.

### 2.3 Proposed Method

Our approach comprises two primary stages: 3D model training and 2D model training. Initially, we aimed to develop our model based on state-of-the-art architectures like Swin UNETR [26] or UNETR [9]. Unfortunately, the performance of these 3D transformer-based models did not meet our expectations and even underperformed in comparison to 3D U-Net [4], as detailed in Sec 4.1. Furthermore, these 3D transformer-based models need excessive GPU memory consumption and long running times, making them cost-ineffective to optimize. Consequently, we made the decision to pivot towards solutions rooted in the conventional 2D U-Net [24] architecture. To maximize the utilization of the unlabeled data, we leveraged the trained 3D models to generate pseudo labels for 2D model training.

**3D Training Stage.** The training process of our 3D models is illustrated in Fig. 1(b). For transformer-based networks, specifically Swin UNETR and UNETR, we initiated the training by pre-training their transformer encoders using the MAE method, as described in [10,3], on the complete training dataset $D$, then we fixed the parameters of the transformer encoders throughout the subsequent steps.

Given that our image data was partially labeled, we pursued the training of two separate models—one for predicting cancer labels and the other for organ labels. These two training processes occurred in parallel. Building upon the pre-trained encoder, one model underwent fine-tuning solely on the dataset $D_c$ to yield the cancer model $M_c$. Concurrently, another model was initially fine-tuned on the dataset $D_{o5}$ to obtain model $M_{o1}$, which was designed to predict labels for five specific organs. Subsequently, model $M_{o1}$ underwent continuous fine-tuning on the dataset $D_{o13}$, ultimately yielding the comprehensive organ model $M_{o2}$, which is capable of predicting labels for all organs.

In contrast, the 3D U-Net did not undergo a pre-training phase or training on dataset $D_{o5}$. Instead, we directly trained two distinct models, $M_c$ and $M_{o2}$, for the segmentation of cancer and organs by utilizing the datasets $D_c$ and $D_{o13}$, respectively.

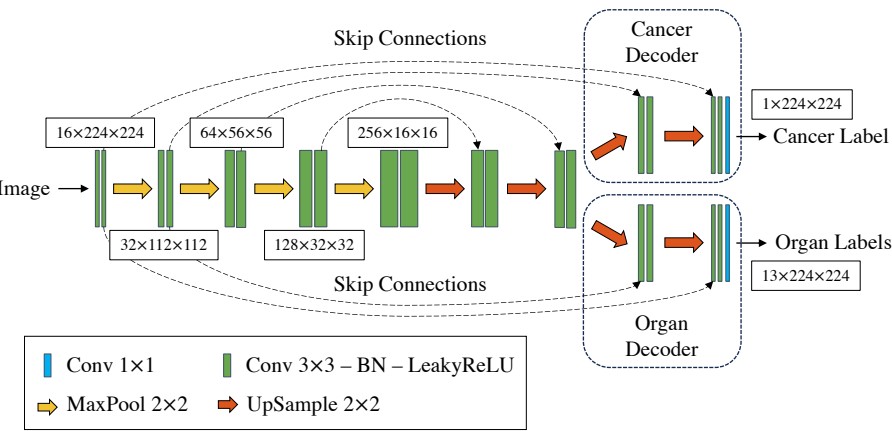

**Fig. 2.** The network architecture of our proposed dual-decoding U-Net. In this configuration, the decoder comprises two branches in the final two upsampling stages, enabling the network to simultaneously train on data labeled for both organs and cancers, and generate corresponding labels for both.

**2D Training Stage.** In light of the insights gained from the training of our 3D models and with the aim of further enhancing efficiency, we have transitioned away from training separate models for cancer and organs. Instead, we have devised a network featuring a dual-decoding structure based on the 2D U-Net

architecture. This design enables the model to produce labels for both cancer and organs concurrently. We refer to the two decoders responsible for generating these labels as the 'cancer decoder' and the 'organ decoder'. A schematic representation of this network is illustrate in Fig. 2. Furthermore, we have ensured that the network's parameters remain unaffected during training on different types of data. This network structure provides the capability to simultaneously extract features related to both cancer and organs from images, while also allowing us to leverage unlabeled data.

To made the use of models $M_c$ and $M_{o1}$ from the 3D U-Net version, we utilized them to predict pseudo labels for all images within $D_{l3}$ and unlabeled images in $D_u$ (Fig. 1(c)). It is important to note that within $D_{l3}$, only pseudo labels for the eight remaining organs were used, as the other five organs already had ground truth labels. For clarity, we denote datasets $D_u$ and $D_{l3}$ with pseudo labels as $D'_u$ and $D'_{l3}$, respectively.

The training process for this 2D model consists of four steps. In the first and second steps, the entire network was trained using datasets $D'_u$ and $D'_{l3}$, respectively. Moving to the third step, training was conducted on dataset $D_{l1}$, which includes thirteen organ labels. The parameters of the network were fixed in this step, except for those of organ decoder. The fourth step is similar to the third step, but dataset $D_{l4}$ was used. In this case, the parameters of the cancer decoder updated, while keeping the other parameters fixed.

**Loss Function.** We used the summation between Dice loss [22] and cross-entropy loss because compound loss functions have been proven to be robust in various medical image segmentation tasks [16]. The loss function is calculated according to the following formula:

$$
\begin{aligned}
L\left(G, P\right) &= L_{dice}\left(G, P\right) + L_{ce}\left(G, P\right) \\
&= \left(1 - \frac{2}{J} \sum_{j=1}^{J} \frac{\sum_{i=1}^{I} g_{i,j} p_{i,j}}{\sum_{i=1}^{I} g_{i,j}^2 + \sum_{i=1}^{I} p_{i,j}^2}\right) + \left(-\frac{1}{I} \sum_{i=1}^{I} \sum_{j=1}^{J} g_{i,j} \log p_{i,j}\right)
\end{aligned} \tag{1}
$$

where $I$ denotes the total number of pixels, $J$ denotes the number of classes. $G$ and $P$ respectively represent the sets of pixels in the ground truth and prediction. For any pixel $g_{i,j} \in G$ and its corresponding prediction $p_{i,j} \in P$, $g_{i,j} = 1$ if the $i$-th pixel is classified into the $j$-th class and $g_{i,j} = 0$ if not, $p_{i,j}$ is the predicted probability of the $i$-th pixel belonging to class $j$.

### 2.4   Post-processing

In our method, whether 3D or 2D models, the initial model outputs provide either cancer labels or organ labels which do not encompass all fourteen labels. The comprehensive predicted results, including all fourteen labels, were achieved by overlaying the cancer labels onto the organ labels.

To ensure that the final prediction results align with the original input image, the inverse transforms of the orientation and spacing during the pre-processing

were performed on the final outputs. Besides, no additional post-processing was applied.

## 3   Experiments

### 3.1   Dataset and Evaluation Measures

The FLARE 2023 challenge is an extension of the FLARE 2021-2022 [19][20], aiming to aim to promote the development of foundation models in abdominal disease analysis. The segmentation targets cover 13 organs and various abdominal lesions. The training dataset is curated from more than 30 medical centers under the license permission, including TCIA [5], LiTS [2], MSD [25], KiTS [11,12], autoPET [7,6], TotalSegmentator [28], and AbdomenCT-1K [21]. The training set includes 4000 abdomen CT scans where 2200 CT scans with partial labels and 1800 CT scans without labels. The validation and testing sets include 100 and 400 CT scans, respectively, which cover various abdominal cancer types, such as liver cancer, kidney cancer, pancreas cancer, colon cancer, gastric cancer, and so on. The organ annotation process used ITK-SNAP [29], nnU-Net [14], and MedSAM [18].

The evaluation metrics encompass two accuracy measures—Dice Similarity Coefficient (DSC) and Normalized Surface Dice (NSD)—alongside two efficiency measures—running time and area under the GPU memory-time curve. These metrics collectively contribute to the ranking computation. Furthermore, the running time and GPU memory consumption are considered within tolerances of 15 seconds and 4 GB, respectively.

### 3.2   Implementation Details

**Table 1.** Development environments and requirements.

| | |
|---|---|
| System | Ubuntu 20.04.5 LTS |
| CPU | Intel(R) Xeon(R) Platinum 8260 CPU @ 2.40GHz |
| RAM | 4 × 16 DDR4 2933 MHz |
| GPU (number and type) | Two NVIDIA V100 32G |
| CUDA version | 12.0 |
| Programming language | Python 3.8.13 |
| Deep learning framework | torch 1.13.0, monai 1.1.0 |
| Specific dependencies | NA |
| Code | NA |

Details regarding the development environments and requirements are presented in Table 1. Our method was implemented using PyTorch[4] and MONAI[5].

---

[4] https://pytorch.org/
[5] https://monai.io/

We leveraged modules and functions from MONAI for pre-processing and post-processing during both training and inference.

The non-random preprocessing, as introduced in Sec 2.2, was consistently applied during both training and inference. Additionally, for the purpose of data augmentation during training, several other transforms were employed:

- Crop Foreground: All input images underwent cropping based on their image intensity. Only the largest bounding box containing voxel values greater than 0 within the image was retained.
- Randomly Crop Samples: For 3D models, input images were randomly cropped into two patches of size $128 \times 128 \times 128$. For 2D models, input images were randomly cropped into samples of size $224 \times 224$.
- Gaussian Noise: Gaussian noise with a standard deviation of 0.05 was randomly added with a probability of 0.5.
- Intensity Scaling and Shifting: Image intensities were randomly scaled by a factor of 0.1 and shifted with randomly selected offsets from the range $[-0.1, 0.1]$. Both of these operations occurred with a probability of 0.5.

The training protocols for 3D and 2D U-Net models are outlined in Table 2 and Table 3, respectively. In the case of the 3D U-Net, we selected the model with the lowest Dice loss as the optimal model. As for the 2D U-Net, approximately 20% of the slices were designated as validation data, ensuring that slices from the same image were either included in the validation set or the training set. The optimal model for the 2D U-Net was chosen based on the Dice score achieved on the validation data.

**Table 2.** Training protocols for 3D U-Net.

| Training Model | cancer model $M_c$ | organ model $M_{o2}$ |
|---|---|---|
| Network initialization | random | |
| Batch size | 4 | |
| Patch size | $128 \times 128 \times 128$ | |
| Number of data (cases) | 1492 | 250 |
| Total epochs | 60 | 250 |
| Optimizer | AdamW, weight decay $10^{-5}$ | |
| Initial learning rate (lr) | $5 \times 10^{-4}$ | |
| Lr schedule | warm up of 5 epochs | warm up of 10 epochs |
| Training time | 88 hours | 32 hours |
| Loss function | Dice loss and cross-entropy loss | |
| Number of model parameters | 1.439M | |
| Number of flops | 21.9G | |

**Table 3.** The training protocols for the 2D U-Net models $M_1$ to $M_4$ correspond to the four steps outlined in Fig. 1(d).

| Training Model | $M_1$ | $M_2$ | $M_3$ | $M_4$ |
|---|---|---|---|---|
| Network initialization | random | $M_1$ | $M_2$ | $M_3$ |
| Batch size | 128 | | | |
| Patch size | $224 \times 224$ | | | |
| Number of data (slices) | 330828 | 123698 | 64121 | 349701 |
| Total epochs | 20 | 30 | 60 | 20 |
| Optimizer | AdamW, weight decay $10^{-5}$ | | | |
| Initial learning rate (lr) | $10^{-4}$ | $5 \times 10^{-4}$ | $5 \times 10^{-4}$ | $10^{-4}$ |
| Lr schedule | polynomial decay with power 0.9 | | | |
| Training time | 24.5 hours | 9 hours | 9.5 hours | 26 hours |
| Loss function | Dice loss and cross-entropy loss | | | |
| Number of model parameters | 2.388M | | | |
| Number of flops | 26G | | | |

## 4    Results and Discussion

### 4.1    Quantitative Results on Validation Set

**Results of the Submitted Solution.** The validation results of our submitted solution can be found in Table 4. Regrettably, we did not discover that we had accidentally included the 50 validation cases with ground truth in our training data until writing this paper, despite our initial intent to use this data solely for selecting the optimal model. As a result, the submitted model's performance was unfairly influenced. Consequently, we retrained our models entirely from scratch. The results of the retraining models are presented in subsequent ablation studies, and these results are truly noteworthy and thought-provoking.

**Ablation Studies.** In this part, we present a comparative analysis of the performance of our 3D and 2D models on validation data.

Table 5 displays the online validation results for the 3D models Swin UN-ETR, UNETR, and U-Net. Despite Swin UNETR and UNETR being pre-trained on the entire training dataset, their performance falls significantly short of that achieved by the 3D U-Net model. From the table, it is evident that the average DSC and NSD)scores for the 3D U-Net across all fourteen classes are approximately 4-6% higher compared to the other two models. Regarding cancer segmentation specifically, the DSC score of 3D U-Net surpasses that of Swin UNETR by 0.14% and UNETR by 6.8%, respectively. Furthermore, while Swin UNETR outperforms UNETR marginally in terms of the results, Swin UNETR consumes significantly more GPU memory and takes longer running times during both training and inference when compared to UNETR.

For 2D models, our primary focus was to assess the influence of utilizing unlabeled data on model performance, and the results are presented in Table 6.

**Table 4.** Quantitative evaluation results of our submitted solution. The public validation denotes the performance on the 50 validation cases with ground truth. The online validation denotes the leaderboard results. The Testing results will be released during MICCAI.

| Target | Public Validation | | Online Validation | | Testing | |
|---|---|---|---|---|---|---|
| | DSC(%) | NSD(%) | DSC(%) | NSD(%) | DSC(%) | NSD (%) |
| Liver | 98.45 ± 0.42 | 99.58 ± 0.50 | 97.90 | 98.19 | 95.38 | 94.99 |
| Right Kidney | 97.12 ± 1.74 | 98.67 ± 2.28 | 94.08 | 94.80 | 92.69 | 92.28 |
| Spleen | 97.78 ± 0.62 | 99.13 ± 1.26 | 95.48 | 95.58 | 93.83 | 92.26 |
| Pancreas | 88.32 ± 3.67 | 98.41 ± 1.33 | 79.33 | 91.52 | 72.84 | 85.67 |
| Aorta | 95.87 ± 3.77 | 98.38 ± 2.69 | 94.85 | 96.80 | 94.80 | 96.88 |
| Inferior vena cava | 92.37 ± 3.77 | 94.91 ± 4.35 | 89.61 | 90.58 | 87.77 | 87.991 |
| Right adrenal gland | 82.98 ± 10.86 | 94.70 ± 8.19 | 78.43 | 91.12 | 68.41 | 80.90 |
| Left adrenal gland | 81.62 ± 10.29 | 93.86 ± 7.59 | 73.28 | 84.33 | 64.16 | 74.89 |
| Gallbladder | 91.80 ± 13.78 | 94.10 ± 14.02 | 83.78 | 84.37 | 70.54 | 69.42 |
| Esophagus | 86.09 ± 5.16 | 96.46 ± 3.64 | 81.50 | 92.84 | 79.97 | 90.63 |
| Stomach | 95.38 ± 1.87 | 98.60 ± 1.66 | 91.38 | 92.75 | 84.17 | 83.96 |
| Duodenum | 87.06 ± 5.94 | 99.05 ± 1.20 | 76.00 | 92.31 | 64.53 | 84.15 |
| Left kidney | 96.53 ± 2.27 | 97.64 ± 3.89 | 93.23 | 93.79 | 91.67 | 91.20 |
| Tumor | 87.19 ± 8.30 | 86.22 ± 10.03 | 57.66 | 52.38 | 17.51 | 9.66 |
| Average | 91.33 ± 2.74 | 96.41 ± 3.56 | 84.75 | 89.38 | 76.90 | 80.98 |

**Table 5.** Online validation results of Swin UNETR, UNETR and 3D U-Net.

| Target | Swin UNETR | | UNETR | | 3D U-Net | |
|---|---|---|---|---|---|---|
| | DSC(%) | NSD(%) | DSC(%) | NSD(%) | DSC(%) | NSD (%) |
| Liver | 94.24 | 94.55 | 94.22 | 94.68 | 94.59 | 96.25 |
| Right Kidney | 83.78 | 84.95 | 88.89 | 90.57 | 87.71 | 89.85 |
| Spleen | 91.63 | 92.36 | 89.33 | 89.32 | 90.51 | 91.20 |
| Pancreas | 70.15 | 85.08 | 69.05 | 85.28 | 75.58 | 91.09 |
| Aorta | 86.79 | 88.84 | 86.74 | 86.29 | 92.81 | 95.61 |
| Inferior vena cava | 85.13 | 86.81 | 81.54 | 79.08 | 89.34 | 91.21 |
| Right adrenal gland | 70.23 | 87.18 | 65.12 | 82.13 | 71.46 | 88.48 |
| Left adrenal gland | 61.88 | 77.92 | 60.36 | 75.11 | 68.17 | 85.26 |
| Gallbladder | 69.90 | 66.73 | 64.68 | 57.81 | 70.44 | 66.65 |
| Esophagus | 69.07 | 83.32 | 67.26 | 83.35 | 76.59 | 90.10 |
| Stomach | 82.65 | 84.99 | 78.77 | 80.90 | 86.51 | 90.09 |
| Duodenum | 66.96 | 84.94 | 58.51 | 81.98 | 73.42 | 89.93 |
| Left kidney | 78.41 | 79.65 | 84.72 | 86.27 | 86.68 | 88.79 |
| Tumor | 11.34 | 5.99 | 14.67 | 6.18 | 21.47 | 11.65 |
| Average | 73.01 | 78.81 | 71.70 | 77.07 | 77.52 | 83.30 |

The model that excluded unlabeled data was trained following the steps outlined in Fig. 1(d), omitting step 1. From the table, it is evident that the model's performance, when trained with unlabeled data, does not surpass that of the model trained without unlabeled data. In fact, the overall DSC score has decreased by 2.63%. Notably, the inclusion of unlabeled data resulted in a significant decrease in DSC scores for all organs, except for the liver. This decrease was particu-

**Table 6.** Comparison of quantitative evaluation results in online validation between models trained with and without unlabeled data.

| Model trained | w/o unlabeled data | | with unlabeled data | |
|---|---|---|---|---|
| Target | DSC(%) | NSD(%) | DSC(%) | NSD(%) |
| Liver | 96.67 | 95.62 | 96.88 | 95.19 |
| Right Kidney | 89.24 | 88.3 | 88.72 | 87.69 |
| Spleen | 93.15 | 91.24 | 93.06 | 91.1 |
| Pancreas | 78.54 | 90.39 | 74.63 | 85.99 |
| Aorta | 94.45 | 96.66 | 91.97 | 94.3 |
| Inferior vena cava | 90.41 | 91.65 | 87.46 | 88.05 |
| Right adrenal gland | 69.89 | 84.98 | 68.23 | 80.64 |
| Left adrenal gland | 67.68 | 79.08 | 59.73 | 72.85 |
| Gallbladder | 74.79 | 72.45 | 74.06 | 70.73 |
| Esophagus | 79.05 | 91.35 | 76.07 | 89.01 |
| Stomach | 87.59 | 87.17 | 85.19 | 85.89 |
| Duodenum | 72.72 | 87.95 | 65.12 | 86.24 |
| Left kidney | 87.64 | 86.87 | 83.86 | 83.44 |
| Tumor | 33.22 | 23.32 | 33.31 | 23.1 |
| Average | 79.65 | 83.36 | 77.02 | 81.02 |

larly pronounced in the left adrenal gland and duodenum. However, it is worth mentioning that unlabeled data did lead to a slight improvement in cancer segmentation, with online validation DSC scores 0.11% higher than those achieved by models trained exclusively on labeled data. This performance disparity may be attributed to the absence of filtering or restrictions applied to the pseudo labels generated from unlabeled data. It is worth noting that the 3D U-Net model used for generating these pseudo labels demonstrated limited reliability, as evidenced by its average DSC score, which is only 77.52% in online validation. Consequently, during the final training steps with labeled data, where most model parameters were fixed and only certain parameters in decoder were updated, the model's performance was significantly influenced by the data quality used in the initial training steps. Additionally, the unlabeled data comprises a portion of full-body CT scans, while the validation set exclusively consists of abdominal CT scans. This divergence in data distribution can also contribute to the decline in model performance when unlabeled data was utilized. In the context of cancer segmentation, the considerable variation in the number, size, and location of cancer lesions compared to organs is critical. Therefore, even if the pseudo labels for unlabeled data are not highly accurate, a substantial volume of anisotropic data may still contribute to improving the model's ability to segment cancer to some extent.

### 4.2   Qualitative Results on Validation Set

Fig. 3 presents four segmentation results from the validation set. Specifically, Case ♯00053 and ♯0038 have relatively high DSC scores, while Case ♯0067 and ♯0021 have relatively low DSC scores. The yellow boxes in the first three lines

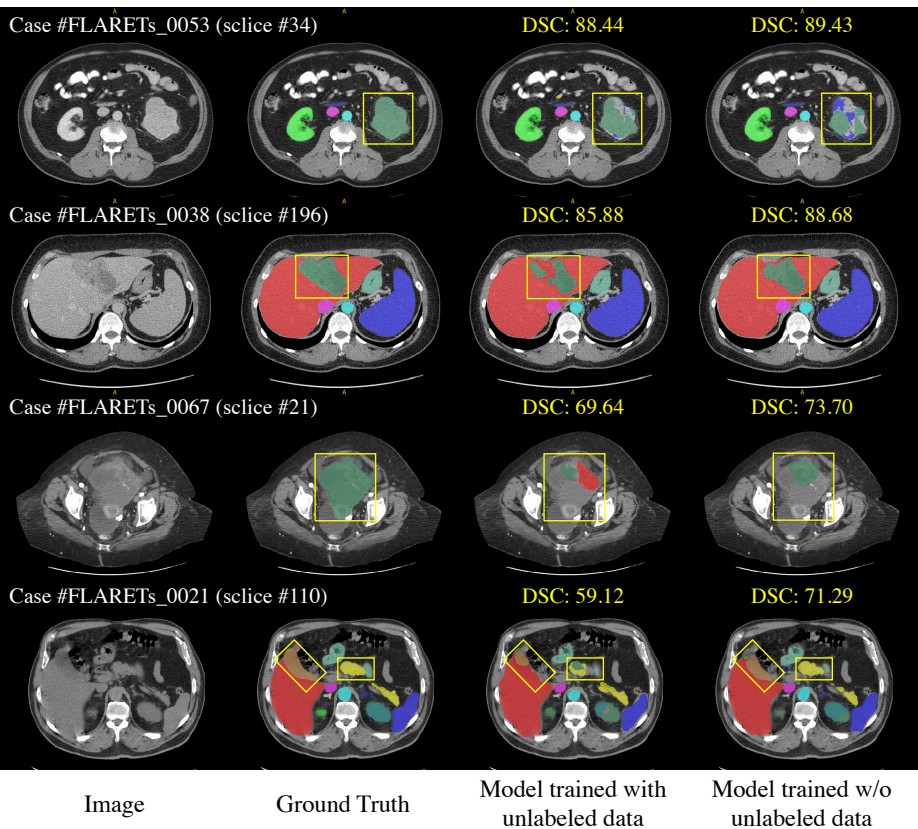

**Fig. 3.** Visualizations of segmentation results. The top two rows showcase two examples with good segmentation results, while the bottom two rows display two instances with bad segmentation results in the validation set. The DSC scores (%) are calculated for cases, not for slices. Areas with notable differences between the model's segmentation and ground truth have been highlighted within yellow boxes.

signify that the trained model encounters difficulties in cancer segmentation. When cancer lesions are large or significantly alter the original organ shape, achieving accurate segmentation becomes a challenging task. In addition, these four examples collectively illustrate that models trained with unlabeled data exhibit greater stability in organ segmentation compared to models trained without unlabeled data. This finding aligns with the conclusion presented in Table 6.

### 4.3   Segmentation Efficiency Results on Validation Set

Though our submitted model did not achieve fairness in validation results, it is essential to note that its network structure remained unchanged in the retrained model, ensuring the validity and consistency of segmentation efficiency results.

**Table 7.** Quantitative evaluation of segmentation efficiency in terms of the running time and GPU memory consumption. Total GPU denotes the area under GPU Memory-Time curve. Evaluation GPU platform: NVIDIA QUADRO RTX5000 (16G).

| Case ID | Image Size | Running Time (s) | Max GPU (MB) | Total GPU (MB) |
|---------|------------|------------------|--------------|----------------|
| 0001 | (512, 512, 55) | 50.82 | 3012 | 41724 |
| 0051 | (512, 512, 100) | 34.39 | 3860 | 53855 |
| 0017 | (512, 512, 150) | 48.45 | 4642 | 56855 |
| 0019 | (512, 512, 215) | 42.4 | 5360 | 42851 |
| 0099 | (512, 512, 334) | 38.16 | 7152 | 73287 |
| 0063 | (512, 512, 448) | 33.82 | 8796 | 72601 |
| 0048 | (512, 512, 499) | 35.92 | 9492 | 78669 |
| 0029 | (512, 512, 554) | 52.49 | 10424 | 121664 |

Table 7 presents the segmentation efficiency for eight validation cases, arranged in order of increasing image depths from top to bottom. Notably, GPU and total memory consumption increased as the number of image layers increased, with only Case ♯0001 and Case ♯0051 utilizing the maximum GPU memory within the recommended 4GB limit. Additionally, the running times do not exhibit consistent changes with varying image sizes. This discrepancy arises because all images were reshaped to have identical spacing before prediction, resulting in inconsistent numbers of slices to be inferred compared to the original images.

It's important to mention that in post-processing, our initial approach was to resize the predicted labels using nearest-neighbor interpolation, an operation suitable for CPU. However, we faced a challenge that the orientations of images were not consistently aligned. We had uniformly applied MONAI modules to adjust the orientation during pre-processing, and only MONAI's inverse modules were capable of reversing this transformation. Since both pre- and post-processing operations in MONAI rely on GPU, this approach resulted in nearly doubling the GPU memory consumption.

### 4.4   Results on Final Testing Set

This is a placeholder. We will send you the testing results during MICCAI (2023.10.8).

### 4.5   Limitation and Future Work

Cancer segmentation remains a persistent challenge in our research, with results that continue to fall short of expectations. The intricacies of training high-performing models on datasets characterized by inconsistent annotations and anisotropic images represent a compelling and enduring topic in the field of medical image analysis. Further exploration and innovation in this domain are warranted to advance the state-of-the-art and improve the accuracy of cancer segmentation together with organ segmentation.

In the analysis in Sec 4.1, we observed that the quality of pseudo labels from unlabeled data could influence model performance in our approach. Although we were provided with pseudo labels generated by the FLARE22 winning algorithm [13] and the best-accuracy-algorithm [27], we did not incorporate them into our work. Additionally, our method comprises multiple training steps, yet our ablation analysis solely focused on the utilization of unlabeled data. Furthermore, we did not conduct ablation experiments to assess the impact of our designed dual-decoding network framework. As a result, future research can also explore the effects of varying pseudo label quality on model performance. Moreover, investigations can be extended to evaluate the influence of different methodological steps and network structures on model performance. This comprehensive analysis can provide a deeper understanding of the factors contributing to model effectiveness.

## 5   Conclusion

In the FLARE2023 challenge, we presented a cascaded dual-decoding U-Net solution for this partial-labeled abdominal organ and cancer segmentation. Throughout our research, we explored various model architectures, including both transformer-based and CNN-based models, as well as 3D and 2D models. Through continuous analysis of results and strategy adjustments, we ultimately adopted a designed 2D dual-coding U-Net, utilizing 3D U-Net for pseudo label generation and conducting multi-step iterative training. We also conducted an in-depth analysis of the influence of unlabeled data on model performance. Interestingly, our findings demonstrated that pseudo labels of low quality may not only fail to improve model performance but can even degrade the model's organ segmentation performance, as indicated by the results of online validation. In conclusion, our model trained without the use of unlabeled data achieved average DSC and NSD scores of 79.65% and 83.36%, respectively, in online validation.

**Acknowledgements** The authors of this paper declare that the segmentation method they implemented for participation in the FLARE 2023 challenge has not used any pre-trained models nor additional datasets other than those provided by the organizers. The proposed solution is fully automatic without any manual intervention. We thank all the data owners for making the CT scans publicly available and CodaLab [23] for hosting the challenge platform. This work is supported in part by the Major Key Project of PCL (grant No. PCL2023AS7-1) and the National Natural Science Foundation of China (grant No. U21A20523). The computing resources of Pengcheng Cloudbrain are used in this research. We acknowledge the support provided by OpenI Community[6].

---

[6] https://git.openi.org.cn

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

**Table 8.** Checklist Table. Please fill out this checklist table in the answer column.

| Requirements | Answer |
| --- | --- |
| A meaningful title | Yes |
| The number of authors ($\leq 6$) | 3 |
| Author affiliations, Email, and ORCID | Yes |
| Corresponding author is marked | Yes |
| Validation scores are presented in the abstract | Yes |
| Introduction includes at least three parts: background, related work, and motivation | Yes |
| A pipeline/network figure is provided | Fig. 1 & 2 |
| Pre-processing | Page 5 |
| Strategies to use the partial label | Page 5-7 |
| Strategies to use the unlabeled images | Page 5-7 |
| Strategies to improve model inference | Page 6-7 |
| Post-processing | Page 7 |
| Dataset and evaluation metric section is presented | Page 7-8 |
| Environment setting table is provided | Table 1 |
| Training protocol table is provided | Table 2 & 3 |
| Ablation study | Page 10-12 |
| Efficiency evaluation results are provided | Table 7 |
| Visualized segmentation example is provided | Figure 3 |
| Limitation and future work are presented | Yes |
| Reference format is consistent | Yes |