# OpenReview forum: "Partial-labeled Abdominal Organ and Cancer Segmentation via Cascaded Dual-decoding U-Net"
_MICCAI.org/2023/FLARE — Submitted to FLARE 2023_

### Official Review · Reviewer_5teD · 2023-09-20
**Excellent method design description, but numerical results require correction.**

**Rating:** 7
**Confidence:** 5

**Review:**

# Summary
The paper proposes a "Cascaded Dual-decoding U-Net" framework for partial-labeled abdominal organ and cancer segmentation, initially exploring 3D transformer-based models and later transitioning to 2D U-Net solutions due to computational constraints.

# Strengths
1.The paper is well-written and comprehensive, especially with beautifully crafted figures.
2.The authors provide a detailed account of their methodological choices during the competition and the reasons behind those decisions.

# Weaknesses
1.Despite acknowledging the inadvertent inclusion of validation cases in their training data, the authors could have recalculated and updated the results in Table 4.
2.It would be beneficial if the code were made open-source.

---

> ### Author Response · Authors · 2023-11-10
> **Reply towards reviewer 5teD's comments**
>
> Thank you very much for your careful review and constructive comments on this paper. We have carefully considered your suggestions and will address your questions point by point.
>
> 1. Despite acknowledging the inadvertent inclusion of validation cases in their training data, the authors could have recalculated and updated the results in Table 4.
>
> We acknowledge that this issue arose due to our oversight during the training process. Since the test docker submission had already been made by the time we identified the problem, the results in Table 4 were unalterable. Table 6 presents the outcomes after retraining, following the removal of 50 validation set data, which enables us to conduct an objective analysis of our method.
>
> 2. It would be beneficial if the code were made open-source.
>
> Thank you for your suggestion. We have open-sourced the code to https://openi.pcl.ac.cn/OpenMedIA/pclmedia_FLARE23, please check.

---

### Official Review · Reviewer_uKrA · 2023-09-27
**Partial-labeled Abdominal Organ and Cancer Segmentation via Cascaded Dual-decoding U-Net**

**Rating:** 7
**Confidence:** 5

**Review:**

Strengths:
1. The pre-processing strategies for organs and tumors are simple and effective.
2. Conduct qualitative analysis and division of the dataset, and formulate corresponding training strategies.
3. The paper is generally well-written and easy to follow.
4. This paper has made various attempts based on transformer and CNN models, and tried to introduce self-supervised strategies into segmentation tasks.

Weaknesses:
1. The complexity of the training strategy is extremely high. The authors did not comment on this limitation.
2. There are loopholes in the quantitative results analysis in this paper: the author added 50 validation set data to the model training.
3. The paper does not attempt to incorporate pseudo-tags provided by the organizer

---

> ### Author Response · Authors · 2023-11-10
> **Reply towards reviewer uKrA's comments**
>
> Thank you very much for your careful review and constructive comments on this paper. We have carefully considered your suggestions and will address your questions point by point:
>
> 1. The complexity of the training strategy is extremely high. The authors did not comment on this limitation.
>
> We appreciate the reviewer's insightful comment on the complexity of our training strategy. The complexity arises primarily from the need to handle the partial-labeled abdominal organ and cancer segmentation dataset, which is known for its diverse data distribution and  high labelling variability. To mitigate the perceived complexity, we have made the training pipeline fully automatic and open-sourced to enhance the transparency of our methodology. We will explore the simplification without compromising the model's efficacy in future.
>
> 2. There are loopholes in the quantitative results analysis in this paper: the author added 50 validation set data to the model training.
>
> We acknowledge that this issue arose due to our oversight during the training process. Since the test docker submission had already been made by the time we identified the problem, the results in Table 4 were unalterable. Table 6 presents the outcomes after retraining, following the removal of 50 validation set data, which enables us to conduct an objective analysis of our method.
>
> 3. The paper does not attempt to incorporate pseudo-tags provided by the organizer
>
> We chose to use our own model to generated pseudo labels instead of those provided by the organizer in this work. Speculating that utilizing the provided pseudo labels may yield better results, which could be explored in future research.

---

### Official Review · Reviewer_x8gV · 2023-10-03
**Well-designed method, clear written with minor issues**

**Rating:** 8
**Confidence:** 5

**Review:**

The authors propose a two-stage method for training with the partially label training images for abdominal organ and cancer segmentation. The method is clearly described, and the overall paper is well written. Some minor issues exist in the current version:

1) It would be good to add an overall tittle for figure 1.
2) Some details about the dataset are missing. For example, what was the range of resolution in the dataset?
3) For the training losses, please add a definition of Dice and cross entropy loss.
4) After using the additional unlabeled images for training, the performance was decreased. Can you analyze why?
5) The method used MAE-based pre-training, but the effectiveness of pretraining was not shown. What was the difference between using and not using pretraining?

---

> ### Author Response · Authors · 2023-11-10
> **Reply towards reviewer x8gV's comments**
>
> Thank you very much for your careful review and constructive comments on this paper. We have carefully considered your suggestions and will address your questions point by point:
> 1. It would be good to add an overall tittle for figure 1.
>
> We have added "Visualization of our data partitioning method and training strategy" as the overall title of Figure 1.
>
> 2. Some details about the dataset are missing. For example, what was the range of resolution in the dataset?
>
> In our method, we first standardize the data by adjusting the spacing and cropping before training and inference. The parameter settings for data preprocessing, such as image intensity, spacing, and crop size, are typically applied to abdominal CT images and were not specifically customized for the provided dataset. Therefore, including detailed dataset information in the paper is not necessary.
>
> 3. For the training losses, please add a definition of Dice and cross entropy loss.
>
> We used the sum of dice loss and ce loss as our loss function, and the detailed calculation formula has been added to Section "Loss Function" on page 7.
>
> 4. After using the additional unlabeled images for training, the performance was decreased. Can you analyze why?
>
> As the analysis on page 12 (below Table 6) of the paper, our 2D model heavily relies on pseudo labels during the training process, thus, the quality of pseudo labels significantly impacts the model's performance. Additionally, the unlabeled data comprises a portion of full-body CT scans, while the validation set exclusively consists of abdominal CT scans. This divergence in data distribution can also contribute to the decline in model performance when unlabeled data was utilized.
>
> 5. The method used MAE-based pre-training, but the effectiveness of pretraining was not shown. What was the difference between using and not using pretraining?
>
> The MAE method was used to develop our 3D transformer-based models in the initial attempt, as the unlabeled data can be utilized by self-training. However, due to the limitations of computing resources, we decoded to use 2D model instead of 3D model without any pretraining. Therefore the differnece between using and not using pretraining was not invesigated in this work.

---

### Decision · Program_Chairs · 2023-10-24

Accept